Present and future ecological niche modeling of garter snake species from the Trans-Mexican Volcanic Belt

González-Fernández Andrea andreagofe@gmail.com 1
Manjarrez Javier jsilva@ecologia.unam.mx 1
García-Vázquez Uri 2
D’Addario Maristella 3
Sunny Armando sunny.biologia@gmail.com 3
1 Facultad de Ciencias, Universidad Autónoma del Estado de México , Toluca , Estado de México , México
2 Facultad de Estudios Superiores Zaragoza, Universidad Nacional Autónoma de México , Ciudad de México , México
3 Centro de Investigación en Ciencias Biológicas Aplicadas, Universidad Autónoma del Estado de México , Toluca , Estado de México , México
Cardoso da Silva Jose Maria
Electronic publication date: 2018 Apr 11
Publication date: 2018
Volume: 6
Electronic Location ID: e4618
Received 2017 Dec 15; Accepted 2018 Mar 24
Copyright: ©2018 González-Fernández et al.
Copyright year: 2018
Copyright holder: González-Fernández et al.
License: This is an open access article distributed under the terms of the Creative Commons Attribution License, which permits unrestricted use, distribution, reproduction and adaptation in any medium and for any purpose provided that it is properly attributed. For attribution, the original author(s), title, publication source (PeerJ) and either DOI or URL of the article must be cited.
License URL: https://creativecommons.org/licenses/by/4.0/

Keywords: Climate change, Environmental niche models, Thamnophis, Potential distribution, Land-use change

Funding: Universidad Autónoma del Estado de México 4047/2016SF CONACYT UAEMEX This study was supported by a research grant from the Universidad Autónoma del Estado de México (4047/2016SF), and Andrea González-Fernández received scholarships from CONACYT and UAEMEX. The funders had no role in study design, data collection and analysis, decision to publish, or preparation of the manuscript.

==============================
Land use and climate change are affecting the abundance and distribution of species. The Trans-Mexican Volcanic Belt (TMVB) is a very diverse region due to geological history, geographic position, and climate. It is also one of the most disturbed regions in Mexico. Reptiles are particularly sensitive to environmental changes due to their low dispersal capacity and thermal ecology. In this study, we define the important environmental variables (considering climate, topography, and land use) and potential distribution (present and future) of the five Thamnophis species present in TMVB. To do so, we used the maximum entropy modeling software (MAXENT). First, we modeled to select the most important variables to explain the distribution of each species, then we modeled again using only the most important variables and projected these models to the future considering a middle-moderate climate change scenario (rcp45), and land use and vegetation variables for the year 2050 (generated according to land use changes that occurred between years 2002 and 2011). Arid vegetation had an important negative effect on habitat suitability for all species, and minimum temperature of the coldest month was important for four of the five species. Thamnophis cyrtopsis was the species with the lowest tolerance to minimum temperatures. The maximum temperature of the warmest month was important for T. scalaris and T. cyrtopsis. Low percentages of agriculture were positive for T. eques and T. melanogaster but, at higher values, agriculture had a negative effect on habitat suitability for both species. Elevation was the most important variable to explain T. eques and T. melanogaster potential distribution while distance to Abies forests was the most important variable for T. scalaris and T. scaliger. All species had a high proportion of their potential distribution in the TMVB. However, according to our models, all Thamnophis species will experience reductions in their potential distribution in this region. T. scalaris will suffer the biggest reduction because this species is limited by high temperatures and will not be able to shift its distribution upward, as it is already present in the highest elevations of the TMVB.

Introduction

Land use and climate change are affecting the abundance and distribution of species, altering biological communities, ecosystems, and their associated services to humans (Parmesan & Yohe, 2003; Cardinale et al., 2012; Kortsch et al., 2015; Nadeau, Urban & Bridle, 2017). Both factors are the main contributors to the global decline of reptiles (Ribeiro et al., 2009; Schneider-Maunoury et al., 2016; Sunny, González-Fernández & D’Addario, 2017). In fact, some studies indicate that 20% of the world’s reptile species are threatened (Böhm et al., 2013) because they are particularly sensitive to environmental changes due to their low dispersal capacity and thermal ecology (Huey, 1982; Castellano & Valone, 2006; Ribeiro et al., 2009; Russildi et al., 2016). Studies predicting biological responses to land use and climate change are therefore necessary to assess the potential impacts of these changes and develop management decisions and conservation strategies (Jimenez-Valverde & Lobo, 2007; Nadeau, Urban & Bridle, 2017) to mitigate negative impacts. Information concerning species’ distributions is essential in these cases (Liu, White & Newell, 2013). Through species occurrence data and environmental information, we can generate environmental niche models that can be used to predict the location of particular areas where environmental conditions are favorable for the presence of the species of study (Suárez-Atilano, 2015).

The Trans-Mexican Volcanic Belt (TMVB) is a set of mountain ranges and volcanoes of different ages, aligned on a strip that crosses Mexico from west to east. It is a transition area between Nearctic and Neotropical regions that results in an overlap of biotas from both regions (Suárez-Atilano, 2015). Its geological history and geographic position make it a very complex area with 30 distinct climatic types and several different vegetation communities, such as coniferous forests (Pinus sp. and Abies sp.), oak forests (Quercus sp.), mesophyll forests, alpine pastures, subalpine scrub, and riparian vegetation zones (Espinoza & Ocegueda, 2007). For these reasons, the TMVB has the second highest herpetological richness in Mexico and is the most important biogeographic region of the country in number of endemic amphibian and reptile species (Flores-Villela & Canseco-Márquez, 2007; Sunny, González-Fernández & D’Addario, 2017). Due to the complex characteristics of the TMVB, the montane taxa of this region have been exposed to a sky-island dynamic through climate fluctuations (Mastretta-Yanes et al., 2015), consequently, the high-altitude-adapted species could be especially vulnerable to climate change as they may be limited by future rising temperatures (Sunny, González-Fernández & D’Addario, 2017). Moreover, the TMVB is one of the most disturbed regions in the country as it contains the biggest metropolitan areas of Mexico (CONAPO, 2010; Sunny, González-Fernández & D’Addario,  2017).

Garter snakes are among the most abundant snake species in North America (Rossman, Ford & Seigel, 1996; De Queiroz, Lawson & Lemos-Espinal, 2002) and they are distributed from Canada to Costa Rica (Rossman, Ford & Seigel, 1996; Manjarrez, 1998; De Queiroz, Lawson & Lemos-Espinal, 2002). However, we lack information on the ecology and current conservation status of most Thamnophis species that are endemic to, or primarily distributed in, Mexico (Manjarrez, Venegas-Barrera & García-Guadarrama, 2007). They are also the most abundant snake genus in the TMVB (Flores-Villela, Canseco-Márquez & Ochoa-Ochoa, 2010), thus they have an important ecological role in the ecosystem (Montoya, Pimm & Solé, 2006). These garter snakes also have great ecological plasticity in reproduction, feeding, and thermal ecology (Seigel, 1996).

For this study, we chose the five Thamnophis species that occur in the TMVB. Thamnophis melanogaster is endemic to the Mexican Central Plateau. It is a semiaquatic species that inhabits the edges of water bodies and specializes in underwater foraging, preying on aquatic animals such as fish, tadpoles and leeches (Rossman, Ford & Seigel, 1996). Thamnophis scalaris is endemic to high elevations across the TMVB (Rossman, Ford & Seigel, 1996). It lives in grasslands and the periphery of forests, and feeds mainly on earthworms, although it can eat vertebrates such as mice and lizards (Uribe-Peña, Ramirez-Bautista & Casas-Andreu, 1999; Manjarrez, Venegas-Barrera & García-Guadarrama, 2007). Thamnophis scaliger is a poorly-known montane species, endemic to central Mexico (Rossman, Ford & Seigel, 1996). It inhabits forests where it feeds on frogs, salamanders and lizards (Uribe-Peña, Ramirez-Bautista & Casas-Andreu, 1999). Thamnophis cyrtopsis extends from the southern United States to Guatemala, although is mainly distributed in Mexico (Hammerson, 2013). It is an amphibian specialist in aquatic habitats from subtropical deciduous and mixed forests (Rossman, Ford & Seigel, 1996). Thamnophis eques is widely distributed over the Mexican Plateau, reaching southern Arizona and New Mexico (Rossman, Ford & Seigel, 1996). It is a generalist predator because it feeds on both aquatic and terrestrial prey—mostly frogs, tadpoles, and fish, supplemented by lizards and mice (Drummond & Macías García, 1989; Manjarrez, 1998).

Despite their widespread distribution and relatively high abundance in comparison to other reptiles, this group has suffered critical reductions in the last 10 years (Canseco-Márquez & Mendoza-Quijano, 2007; Hammerson, 2013; Vázquez Díaz & Quintero Díaz, 2007; Hammerson, Vázquez Díaz & Quintero Díaz, 2007). Thus, knowledge regarding their ecological niche and their present and future potential distribution is key to better understanding the causes of their population decline. Moreover, due to the geological history and geographic position of the TMVB, the animals of this region constitute a cenocron (a group of animals originated in a defined area that have coexisted for a long period, thus sharing a common biogeographic history and a distribution pattern; Halffter & Morrone, 2017), therefore, changes in garter snake species’ distributions in the TMVB may represent future changes in other species’ distributions of this region. We expect that land use and climate change will reduce the future potential distribution of these five garter snake species (i.e., a reduction in the suitable area available for each species). In this study, we aimed to answer the following four questions. (1) Which climatic, topographic and land use variables determine each species’ distribution? (2) Considering land use and climate change, what are the present and future potential distributions of each species? (3) What changes in suitable available area in the TMVB will each species undergo in the future? (4) What changes in suitable available area in the entire country will each species undergo in the future?

Materials and Methods

We modeled the potential distribution of the five Thamnophis species that occur in the TMVB (T. cyrtopsis, T. eques, T. melanogaster, T. scalaris, and T. scaliger). Occurrence records were obtained from fieldwork (60% or more; Table S1) and online databases such as Global Biodiversity Information Facility (GBIF) and iNaturalist. We selected only the records from the last 20 years for the analysis, as extensive land use changes occurred in Mexico during the 1990s (FAO, 1993) and we included some land use variables (such as extent of induced grasslands and agriculture areas) in the analysis. Maps of occurrence data for all five species were generated to check for obvious errors. We also filtered these data to eliminate duplicated observations from the same pixel (1-km resolution). We defined a polygon (background) for each species that represents the accessibility area (Suárez-Atilano, Burbrink & Vázquez-Domínguez, 2014; Suárez-Atilano et al., 2017). These polygons were generated considering biogeographic regions with geographical records or records near their borders (Sunny, González-Fernández & D’Addario, 2017). Three of the species are endemic to Mexico and the other two have only a marginal distribution outside the country (Rossman, Ford & Seigel, 1996; Hammerson, 2013), therefore all polygons are large representative regions of species distribution ranges. We obtained bioclimatic variables from WorldClim (Hijmans et al., 2005); topographic and land cover variables were obtained from the National Institute of Statistics and Geography (INEGI, 2013). We reclassified the land use map (series V, 1:250,000, generated during the period 2011 to 2013; INEGI, 2013) in different exclusive classes that were converted to raster and transformed from categorical to continuous using a resample method that averages the value of the surrounding pixels to assign a new value to each pixel. All layers were processed in a raster format, with 1-km resolution, using ARC GIS 10.5 and the packages RASTER (Hijmans, 2016) and RGDAL (Bivand, Keitt & Rowlingson, 2017) for R software (version 3.4.0; R Development Core Team, 2017). After a bibliographic review and Pearson correlation analysis to discard highly correlated variables (R2 > 0.8, Suárez-Atilano, 2015) we selected the following variables: elevation, percent natural grasslands, percent human-induced grasslands, percent arid vegetation, percent Pinus forest, distance to Pinus forest, percent Quercus forest, distance to Quercus forest, percent Abies forest, distance to Abies forest, distance to water sources, percent agriculture, minimum temperature of the coldest month, maximum temperature of the warmest month, precipitation of the wettest month, and precipitation of the driest month.

We used the maximum entropy modeling software (MAXENT; Phillips, Anderson & Schapire, 2006) which estimates species’ distributions by finding the distribution of maximum entropy (the most spread out, or closest to uniform), subject to constraints imposed by a known distribution of the species, and by the environmental conditions across the study area (Anderson & González Jr, 2011). First, we ran the model for each species in MAXENT with 10 replicates and we selected the most important variables that explained the distribution of each species (Anderson, Lew & Peterson, 2003; Chefaoui, Hortal & Lobo, 2005; Suárez-Atilano et al., 2017). We only used linear and quadratic features because we had less than 80 records of T. scaliger (Merow, Smith & Silander, 2013) and, for an easier comparison, we used the same methodology for all species. All analyses were performed using the logistic output for an easier interpretation and a convergence threshold of 1 × 10−5 with 500 iterations (Pearson, 2007; Suárez-Atilano, 2015). We modeled again, this time with only the most important variables for each species (Guisan & Zimmerman, 2000; Guisan & Thuiller, 2005; Araujo & Guisan, 2006) and projected these models to the future using both clumping (restricting the variables to the range of values encountered during model calibration) and extrapolation methods (Merow, Smith & Silander, 2013). We obtained the future bioclimatic variables CCSM4 for the year 2050 considering the climate change scenario rcp45 (middle-moderate) from WorldClim. Land use and vegetation variables for the year 2050 were generated using the module LAND CHANGE MODELER FOR ECOLOGICAL SUSTAINABILITY in IDRISI SELVA 17.0 software (Clark Labs, 2012) and land use and vegetation layers from years 2002 and 2011 (series III and V; INEGI, 2005; INEGI, 2013). We also used elevation, slope (obtained from the elevation layer), and distance to urban settlements, for a better prediction of land use change. We designated present urban areas (from the present distribution maps) and future urban areas (from the future distribution maps) as areas of zero habitat suitability. We did not include distance to urban areas as a variable in the models because this can generate a bias, as these areas are more easily accessed by observers ( Araujo & Guisan, 2006). We generated present and future potential distribution maps for each species. We preferred to show the continuous maps because binary outputs can obscure important biological detail (Liu, White & Newell, 2013). To evaluate model performance, we used partial Receiving Operating Characteristic (partial ROC) analyses (Peterson, Papes & Soberón, 2008; Osorio-Olvera, 2016) as recommended based on criticisms of area under the curve analyses (AUC) (Lobo, Jiménez-Valverde & Real, 2008; Peterson, Papes & Soberón, 2008). While AUC evaluates only the environmental niche model (under the omission-commission framework) performance, partial-ROC allows for statistical significance from the AUC itself, based on a null distribution of expectations created via bootstrapping replacement of 50% of the total available points and 1,000 resampling replicates (Suárez-Atilano, 2015). One-tailed significance of the difference between AUC and the null expectations was assessed by fitting a standard normal variate (the z-statistic) and calculating the probability that the mean AUC ratio was ≤1. We used 75% of occurrence localities for model training and 25% for model testing (Suárez-Atilano, 2015). We used the platform NICHE TOOLBOX for partial-ROC calculations (Osorio-Olvera, 2016). We generated the species potential distribution binary maps using Max SS threshold (Liu, White & Newell, 2013), a threshold selection method based on maximizing the sum of sensitivity and specificity. This is considered an adequate method to use when reliable absence data are unavailable (Liu, White & Newell, 2013). For each species, we used these binary maps to calculate the present and future high-suitability areas (Suárez-Atilano, 2015) in both all of Mexico, and the TMVB only, to assess whether the distribution of each species will decrease or increase in the future.

Results

After filtering the data, we worked with 267 records of T. cyrtopsis (Fig. 1A), 274 of T. eques (Fig. 1B), 103 of T. melanogaster (Fig. 1C), 186 of T. scalaris (Fig. 1D), and 76 of T. scaliger (Fig. 1E). The most important variables for each Thamnophis species are summarized in Table 1. In all cases these variables together explained 60% or more of the species’ potential distribution. It is important to note that arid vegetation had an important negative effect on habitat suitability for all species (Fig. S1) and the minimum temperature of the coldest month was important in four of the five models. This latter variable was the most important to explain T. cyrtopsis potential distribution, which was the species with the lowest tolerance to minimum temperatures (5 °C). Habitat suitability for T. scalaris and T. cyrtopsis steadily decreased when maximum temperatures increased. Low agriculture percentages were positive for T. eques and T. melanogaster but, at higher values (above 30%), agriculture had a negative effect on habitat suitability for both species. Elevation was the most important variable to explain T. eques and T. melanogaster potential distribution. It was a positive variable for T. melanogaster, while habitat suitability for T. eques was optimal near 2,500 m above sea level (masl). Distance to Abies forests was the most important variable to explain T. scalaris and T. scaliger potential distribution. It had negative effects on these species (as distance to Abies forests increase, habitat suitability decrease), which means that proximity to Abies forests was positive for both species. Distance to Quercus forests had a negative effect on habitat suitability for T. cyrtopsis, which means that proximity to these forests was positive for the species.

Figure 1 Occurrence records used to build the distribution model for each Thamnophis species, showing the Trans-Mexican Volcanic Belt (TMVB) in dark gray.

Table 1 Contribution percent of the most important variables that explain the distribution of each Thamnophis species.

Variables	T. cyrtopsis	T. eques	T. melanogaster	T. scalaris	T. scaliger	
Minimum temperature of the coldest month	33.7	19.2	11.3		26.5	
Maximum temperature of the warmest month	5.2			36		
Elevation		28.4	27.3			
Arid vegetation	26.4	15.6	11.5	4.9	5.3	
Agriculture		9.6	12.9			
Distance to Quercus forest	8.5					
Distance to Abies forest				44.9	40.6	
Total	73.8	72.8	63	85.8	72.4	

Between 2002 and 2011, there was an increase of almost 16,000 km2 in agriculture and about 5,000 km2 in urban areas (Fig. 2). There was also an increase in human-induced grasslands. A reduction in arid vegetation and natural grasslands occurred, mainly because of its conversion to agriculture lands. The area of Pinus and Quercus forests also fell, but Abies forests held steady. For the year 2050, an increase of 82,865 km2 in agriculture areas is expected according to the model (Figs. 3A, 3B). The urban areas will increase by 20,392 km2 (Figs. 3C 3D), most of it taking place in the area around Toluca city (Figs. 3C, 3D), and induced grasslands will increase by 24,796 km2 (Figs. 3E, 3F). Potential distribution maps for each species are in Figs. 4A–4J. We found no differences in future projections between extrapolation and clumping methods. Partial-ROC bootstrap tests showed significant ratio values of empirical AUC over null expectations (mean AUC ratios ≥1.5 and p-values <0.001 in all cases; Fig. S1).

Figure 2 Land use change by category (km2) between years 2002 and 2011.

Figure 3 Present (2011) and future (2050) maps of (A, B) agriculture, in red; (C, D) urban, in gray; and (E, F) induced grasslands, in orange.

Figure 4 Present (2011) and future (2050) potential distribution maps for each Thamnophis species: (A, B) T. cyrtopsis, (C, D) T. eques, (E, F) T. melanogaster, (G, H) T. scalaris and (I, J) T. scaliger.

All species had a high proportion of their potential distribution in the TMVB. However, according to high-suitability area calculations for present and future, all Thamnophis species will experience reductions in their distribution in this region (Table 2). T. scaliger was only distributed in the TMVB, while T. scalaris had a small part of its potential distribution in the Sierra Madre del Sur. T. cyrtopsis, T. eques, and T. melanogaster were also distributed in the Sierra Madre Occidental, Sierra Madre Oriental, Sierra Madre del Sur, and Oaxaca mountain ranges. Unlike the TMVB, these biogeographic regions will not suffer important reductions in suitable habitat for T. eques and T. melanogaster in the future. T. cyrtopsis will suffer important reductions in all its potential distribution, which also includes Chiapas Highlands. The potential distribution of T. melanogaster will increase in the future, considering the entire country, and T. scalaris will suffer the biggest reduction of the five species (reductions of 54.08% for the TMVB, and 54.30% for all of Mexico, Table 2; Figs. 4A–4J).

Table 2 Present and future high suitability area (km2) and percent of reductions in these areas for each Thamnophis species in Mexico and the Trans-Mexican Volcanic Belt (TMVB).

	MEXICO	TMVB	
	Present distribution (Km2)	Future distribution (Km2)	Reduction (%)	Present distribution (Km2)	Future distribution (Km2)	Reduction (%)	
T. cyrtopsis	661,888.53	387,393.67	41.47	103,190.15	56,172.18	45.56	
T. eques	583,936.04	554,336.36	5.07	102,001.64	88,928.44	12.82	
T. melanogaster	255,647.78	317,411.39	−24.16	83,237.55	67,581.46	18.81	
T. scalaris	110,441.63	50,474.08	54.30	54,057.65	24,825.27	54.08	
T. scaliger	58,682.16	37,278.67	36.47	42,804.76	26,617.94	37.82	

Discussion

Environmental variables

Although current records and literature support the idea that grasslands and water sources are essential for Thamnophis species in Mexico (Jones, 1990; Manjarrez & Drummond, 1996; Venegas-Barrera & Manjarrez, 2011), these variables were not selected by the model as important to explain the distribution of the species. Both variables are more related to the microhabitat of the species, but for this study, we modeled the macrohabitat. Although most records of Thamnophis are in grasslands or near water sources (lakes, ponds and streams), these habitat features are present throughout most of the country, including areas where the species is not present, therefore, these variables are not limiting the species at a macro level. The percent of arid vegetation (which can be interpreted as the opposite of water sources) was a negative limiting factor for all species (Table 1). Distances to forests were more important for explaining the presence of Thamnophis species than the percent of these forests. This was especially important for T. scalaris and T. scaliger as distance to Abies forest was the most important variable determining their potential distribution. These results are consistent with our fieldwork observations, as we found only a few individuals inside forests; the majority were found in grasslands near coniferous forests. This could be because coniferous forests occur in a moist (1,000–3,800 mm annual precipitation) and cold microclimate (2°–24 °C; Sáenz-Romero et al., 2012; Sunny, González-Fernández & D’Addario, 2017) that is preferred by Thamnophis species (Manjarrez & Drummond, 1996). Therefore, microclimatic conditions of grasslands surrounded by forests and large areas of grasslands without forest, may be different. Grasslands surrounded by forests offer the climatic benefits of forests (moist and cold), and the food benefits of grasslands (higher availability of small prey; (Bastos, Araújo & Silva, 2005; Reinert et al., 2011; Wittenberg, 2012; Mociño Deloya, Setser & Pérez-Ramos, 2014).

Low percent of agriculture was positive for T. eques and T. melanogaster, but a high percent was negative for both species. This could be because agriculture is a tradeoff for many reptile species, especially snakes. It provides benefits for them, such as higher prey availability, but also exposes them to human interactions (i.e., people kill Thamnophis out of fear even though these species are not dangerous to humans; Sunny et al., 2015). Moreover, the persistent practice of crop burning and use of roller-chopping to prepare fields also affect their populations (Mullin & Seigel, 2009).

Environmental temperature is important for ectothermic species like garter snakes because they are more active when they can maintain a body temperature above approximately 22 °C (Manjarrez & Drummond, 1996). Environmental temperature increases may lead Thamnophis to physiological stress that results in reduced fitness (Peterson, Gibson & Dorcas, 1993). The fact that T. cyrtopsis was limited by low and high temperatures could be the cause of its reduced potential distribution. The maximum temperature of the warmest month was one of the most important variables explaining T. scalaris potential distribution; however, for all other species, the minimum temperature of the coldest month was more important. This may be because T. scalaris is the species occurring at the highest elevation and, consequently, is adapted to a colder climate. Therefore, while other species are more limited by lower temperatures, T. scalaris is more limited by higher ones, which could make this species more vulnerable to warming temperatures associated with climate change. This scenario is consistent with the future distribution model for this species as T. scalaris suffered the biggest reduction of the five species. The fact that its distribution already includes the existing areas with the highest altitude implies that, as climate change progresses, this species will be limited in its ability to shift its distribution upward, increasing the possibility of becoming extinct (Sunny, González-Fernández & D’Addario, 2017). According to the International Union for Conservation of Nature (IUCN), T. scalaris is considered a species of Least Concern (Canseco-Márquez & Mendoza-Quijano, 2007); however, our results suggest that this risk category is likely to change in the future.

Present and future potential distribution

The potential distribution of all species was located at high elevation areas (mountain ranges), which is consistent with the biology of this genus (Rossman, Ford & Seigel, 1996). The fact that all species had a high proportion of their potential distribution in the TMVB means that this is a very important biogeographic region for the conservation of these five Thamnophis species. This is especially applicable for T. scalaris and T. scaliger as their suitable habitat was mainly found in the TMVB. However, according to our models, all species will suffer large reductions in their potential distribution in the TMVB, while in other regions some species like T. eques and T. melanogaster will not. This may be because the TMVB is one of the most disturbed regions as it contains the largest extent of urban area in Mexico (CONAPO, 2010). Considering the entire country, all species will suffer reductions in their potential distributions in the future except T. melanogaster. We are surprised by this fact, as this species is the most threatened of the five, according to the IUCN (Endangered, Vázquez Díaz & Quintero Díaz, 2007). This species is more aquatic than the others (Manjarrez & Drummond, 1996) and so an approach that considers both macrohabitat and microhabitat variables (such as water source and quality) may be necessary for a better prediction of T. melanogaster potential distribution.

Conclusions and conservation implications

Arid vegetation has an important negative effect on habitat suitability for all species, and the minimum temperature of the coldest month is important for four of the five species. T. cyrtopsis has the lowest tolerance to minimum temperatures. Maximum temperature of the warmest month is important for T. scalaris and T. cyrtopsis. Low percentages of agricultural areas are positive for T. eques and T. melanogaster but at higher values agriculture has a negative effect on habitat suitability for both species. Elevation is the most important variable to explain T. eques and T. melanogaster potential distribution while distance to Abies forests is the most important variable to explain T. scalaris and T.  scaliger potential distribution. As we predicted, all Thamnophis species will experience reductions in their distributions in the TMVB, however, for the entire country, T. melanogaster seems to increase its distribution in the future. We feel more studies should be conducted to evaluate T. melanogaster distribution and abundance. These studies should consider microhabitat variables such as water source and their quality. We also consider it essential to carry out studies of T. scalaris abundance, as this species will suffer the biggest reduction in potential distribution of the five species. Current abundance data of this species will be key to deciding if a change in its conservation status is needed. We are especially concerned about our finding that a relatively abundant species like T. scalaris may suffer severe reductions in its potential distribution, as this suggests that reptile species with similar distributions (such as the lizard Barisia imbricata; Sunny, González-Fernández & D’Addario, 2017) may undergo similar reductions. For less abundant species with similar distributions (such as the rattlesnake Crotalus triseriatus; Sunny et al., 2015), even larger reductions may result. Reductions in suitable area available for Thamnophis and other species will cause the reduction and isolation of their populations. Small populations are susceptible to demographic stochasticity (Gibbs, 1998; Hicks & Pearson, 2003) that can convert normal population fluctuations into local extinctions (Gibbs, 1998). Moreover, while certain isolation levels between populations may facilitate precise evolutionary adaptations to local conditions (Tscharntke et al., 2012), the high isolation levels affecting populations of many species in the TMVB, which are expected to increase in the future and that are limiting species distributions to the highest altitudes of the volcanoes, will lead to important losses of genetic diversity in these populations, thereby affecting their capacity to cope with environmental changes and increasing their susceptibility to extinction (Johansson, Primmer & Merila, 2006; Sunny et al., 2014).

The TMVB has the highest area of Abies forests (91.14%) of the country (Sunny, González-Fernández & D’Addario, 2017); however, it only represents 1.10% of TMBV area (Sunny, González-Fernández & D’Addario, 2017). Unfortunately, governmental laws have recently changed the protection status of some areas of the TMVB, like the Nevado de Toluca Volcano ( DOF, 2013). This change could lead to logging and changes in land use (Mastretta-Yanes et al., 2014). The extent of Abies forests have held steady from 2002 to 2011 (Fig. 2) but we are afraid this could change as a consequence of this new protection status, thus affecting Thamnophis populations and many other species. Moreover, land use changes are expected to accelerate due to climate change (Maclean & Wilson, 2011; Urban, 2015; Nadeau, Urban & Bridle, 2017) so garter snakes and other species of the TMVB could suffer the synergistic effect of both factors. The process of reversing climate change involves world-wide economic systems and government decisions, so there is little we can say about this here. However, we consider the conservation of TMVB forests, especially Abies forests and grasslands associated with them, of great importance for the conservation of many reptile and amphibian species that live in this region (Figueroa-Rangel, Willis & Olvera-Vargas, 2010; Vargas-Rodríguez et al., 2010; Ponce-Reyes et al., 2012; Bryson et al., 2014). Moreover, in the short term, we think is essential to implement environmental education activities to teach everyone the importance of our natural environments. This in turn may lead to fewer reptiles being killed out of fear, and use of fewer wildlife-destructive agricultural practices such as roller chopping and crop burning.

Supplemental Information

Data S1 Geographic coordinates of the Thamnophis species studied

Click here for additional data file.

Supplemental Information 1 Supporting Information

Click here for additional data file.

AGF is grateful to the graduate program “Doctorado en Ciencias Agropecuarias y Recursos Naturales” of the Autonomous University of the State of Mexico and to the Consejo Nacional de Ciencia y Tecnología. We thank Ruthe Smith for valuable comments and English review. We thank the editor and two anonymous reviewers for their comments.

Additional Information and Declarations

Competing Interests

Author Contributions

Data Availability

The authors declare there are no competing interests.

Andrea González-Fernández and Armando Sunny conceived and designed the experiments, performed the experiments, analyzed the data, contributed reagents/materials/analysis tools, prepared figures and/or tables, authored or reviewed drafts of the paper, approved the final draft.

Javier Manjarrez conceived and designed the experiments, contributed reagents/materials/analysis tools, authored or reviewed drafts of the paper, approved the final draft.

Uri García-Vázquez contributed reagents/materials/analysis tools, prepared figures and/or tables, authored or reviewed drafts of the paper, approved the final draft.

Maristella D’Addario conceived and designed the experiments, authored or reviewed drafts of the paper, approved the final draft.

The following information was supplied regarding data availability:

The raw data is provided as a Supplemental File.

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
