# Peer review of "Present and future ecological niche modeling of garter snake species from the Trans-Mexican Volcanic Belt"

_PeerJ, doi:10.7717/peerj.4618_

## Round 0.1 · original submission · Major Revisions

I have completed the review of your manuscript, and a summary is appended below.

Both reviewers recommend reconsideration of your paper following major revision, and I agree with them. I invite you to resubmit your manuscript after addressing all reviewer comments.

I strongly recommend that you provide a better explanation on why you selected five snake species for this study. The referee 1 made excellent suggestions on how to overcome this missing element in your paper.

When resubmitting your manuscript, please carefully consider all issues mentioned in the reviewers' comments, outline every change made point by point, and provide proper rebuttals for any comments not addressed.

Reviewer 1 ·

Basic reporting

The manuscript of Fernández and colleagues (#22530) uses clear and unambiguous, professional English. Overall, the literature provided is appropriate for setting the field background/context. The structure of the manuscript is professional and figures and tables are relevant. The manuscript is clearly self-contained with relevant results to hypotheses. Still, in section General Comments to Authors below, it is listed a series of improvements that should be made to the manuscript.

Experimental design

The manuscript of Fernández and colleagues (#22530) is original primary research that falls within Aims and Scope of the journal. Please see section General Comments to Authors for details on how Research questions and methods should be improved.

Validity of the findings

The conclusions of the manuscript of Fernández and colleagues (#22530) are well stated, linked to original research question, and limited to supporting results. There are no obvious speculations.

Additional comments

The manuscript of Fernández and colleagues addresses the ecological niche modelling of a snake community in a global biodiversity hotspot under climate and habitat change scenarios. The subject is timely and important for the conservation of important biodiversity. Overall, the manuscript is well structured, well written, and main conclusions are well expressed. However, there are multiple aspects that affect the general patterns found. These methodological aspects may have deep effects in the distribution models produced and, thus, serious effects in the predicted patterns of species distribution. Below these aspects are detailed, together with minor details, which need to be addressed in a revision of the manuscript.

L63-L64: "it is the second biogeographic zone with the highest herpetological richness and the most important region in endemic amphibian and reptile species". Which is the scale here used? Is it at global level? Specify in the text the scale of the observation.

L81-L84: The explanation on the why using Thamnophis snakes here should be developed. Given that the TMVB is a biodiversity hotspot, as authors evidenced, then why focusing the study on these 5 snake species? Couldn't the work be developed with a larger faunal assemblage? If not, then authors could explain better the ecological differences among the five species, so that the patterns observed for each one of them may provide indications about expected range shifts in other fauna inhabiting identical ecological conditions. The Introduction could include another paragraph expressing ecological differences and similarities in these five snakes and how representative they may be of other faunal assemblages.

L84-L86: The objectives of the study should be detailed and presented as a list of research questions. These questions should be based on hypotheses that are well explained in the preceding Introduction.

L92: why were selected only observations from the last 20 years?

L93-L94: the information in this sentence is not clear because we need to understand first which is the pixel size to be used in the ecological models. This is information is not given in the manuscript (I could not find it) and it is crucial. Concerning this sentence, if the pixel size of the ecological models is 1x1 km (as I suspect), then this filtering process does not reduce spatial autocorrelation of the presence data, but in fact it eliminates duplicated observation from the same 1x1km pixel. Please revise this sentence for clarity.

L113: which is the pixel size of these variables?

L118-L120: there are important information for the fine-tune of Maxent models, which involves testing distinct parameters (DOI: 10.1111/j.1600-0587.2013.07872.x). Authors should check this publications and apply the guidelines provided.

L122-L124: the models developed in this study are based in partial distributions. In other words, the modelled present range is based in a portion of the ecological conditions where the species range. I assume here that none of them is endemic to the study area; this should be well explain in the Introduction (see above comment for L93). There are abundant literature about the potential effects of modelling partial ranges of species under simulations of future range according to potential climate change impacts (DOI: 10.1111/j.0906-7590.2004.03673.x; DOI: 10.1111/j.1600-0587.2010.06181.x). However, there are conditions when the use of partial range models may be the best option (DOI: 10.1111/ddi.12115). Authors should explain the criteria used to develop ecological models based in partial ranges under climate change scenarios.

L152: replace "see" by "evaluate"

L161-L190: this is a very long section that describes the patterns observed in the figures or tables, species by species. Authors need to make this text more dynamic and interesting. For instance, base this text in the variables and not on the species. Express which are the most important variables, i.e. which variables are more frequently related to the distribution of the five snakes. The same rationality should be applied to figure 1. Authors should combine in the same plot the response curves of multiple species along the same variable. Or alternatively, combine present and future curves for the same species. This way it will be easier to observe differences/similarities among species (in the first option) or across time (second option). Remove from the main text the percentages given in brackets; these values are expressed in the table.

L198-L201: this information should be given in the methods section.

L199: where are the most important variables defined in the manuscript? The methodology for their selection should be given in methods and they should be listed here in Results section

L252: replace "T. Scalaris" by "T. scalaris"

L257-L267: this section is too focused on a exception to the general predictions and not on the "Present and future potential distribution". Authors need to develop this section accordingly.

L306: are the patterns observed in this study anyway representative of any general pattern that could be expected for other fauna inhabiting the region under identical ecological conditions?

FIGURE 3: Include a description of the meaning of the grey colour scale. What is the meaning of the darker and lighter pixels?

Reviewer 2 ·

Basic reporting

I believe the study can be a valuable contribution to the field of niche modeling and to understand how climate changes and land use influence the distribution of suitable habitats for snakes. However, the manuscript needs some crucial improvements. My greatest criticism is related to the way the authors express themselves, mainly when interpreting niche modeling results. I highlighted several examples of this error in the abstract, but they can be found throughout the text. The authors must review these issues carefully before resubmitting the manuscript.

Experimental design

No comment

Validity of the findings

No comment

Annotated reviews are not available for download in order to protect the identity of reviewers who chose to remain anonymous.

---

## Round 0.2 · Major Revisions

I have sent your paper to one of the previous reviewers, and I read the article myself. Although the new version has improved when compared to the first one, there are still several points that require revision. The referee has pointed out the most critical changes that need to be done. Please follow the reviewer's guidance. I invite you to resubmit your manuscript after addressing ALL reviewer's comments.

Although the language is fine in most of the paper, I recommend that the new version of the manuscript needs to be reviewed by a native English speaker to avoid small language mistakes as well as improve the style and the flow of the arguments.

When resubmitting your manuscript, please carefully consider all points mentioned in the reviewers' comments, explain every change made, and provide proper rebuttals for any remarks not addressed.

Reviewer 1 ·

Basic reporting

The manuscript of Fernández and colleagues (#22530) uses clear and unambiguous, professional English. Overall, the literature provided is appropriate for setting the field background/context. The structure of the manuscript is professional and figures and tables are relevant. The manuscript is clearly self-contained with relevant results to hypotheses. Still, below in section General Comments to Authors, it is listed a series of improvements that should be made to the manuscript.

Experimental design

The manuscript of Fernández and colleagues (#22530) is original primary research that falls within Aims and Scope of the journal.

Validity of the findings

The conclusions of the manuscript of Fernández and colleagues (#22530) are well stated, linked to original research question, and limited to supporting results. There are no obvious speculations.

Additional comments

The manuscript of Fernández and colleagues addresses the ecological niche modelling of a snake community in a global biodiversity hotspot under climate and habitat change scenarios. I had the opportunity to revise the initial submission and to provide a series of suggestions to improve the quality of the manuscript. Authors provide a new improved version. However, there are still a few aspects that were not addressed in the review. These are listed below.

L103-L129: It was suggested that authors explain better the ecological similarities/differences among the five Thamnophis species under study and how representative they may be of other faunal assemblages occurring in the region. Authors have included a long text about the ecological traits of the genus, but no reference to the two subjects mentioned above. This section (L103-L129) is now very long and does not reply to the two issues raised before. It should be revised

L130-L134: The objectives of the study should be detailed and presented as a list of research questions. These questions should be based on hypotheses that are well explained in the preceding Introduction.

L214-L239: this text is still very long and mostly describing the patterns observed in the figures or tables. As example, in L217-L218 it is repeated the information (%) given in Table 1. In L223-L226, the text provides a simple description of the plots, which readers can observe by themselves. In L250-L252, the AUC values are repeated from Figure S1. Delete either from main text or from figure S1.
Please do not repeat information given in Tables/Figures in the main text, and vice-versa.

L354-LL364: I acknowledge the effort to turn the predictions for genus Tamnophis comparable to other fauna, but this text (L354-L364) is very long, providing many species names, but without explaining why are the “land-use and climate change in the TMVB” expected to cause changes in these other species. The examples listed in L356-L364 could be reduced to 2 or 3 maximum.

---

## Round 0.3 · accepted · Accept

I read the new version of your paper, and I could see that you followed most the reviewer's suggestions.

#